# COMPOSITIONAL ATTENTION NETWORKS
# FOR MACHINE REASONING

**Drew A. Hudson**
Department of Computer Science
Stanford University
dorarad@cs.stanford.edu

**Christopher D. Manning**
Department of Computer Science
Stanford University
manning@cs.stanford.edu

## ABSTRACT

We present Compositional Attention Networks, a novel fully differentiable neural network architecture, designed to facilitate explicit and expressive reasoning. While many types of neural networks are effective at learning and generalizing from massive quantities of data, this model moves away from monolithic black-box architectures towards a design that provides a strong prior for iterative reasoning, allowing it to support explainable and structured learning, as well as generalization from a modest amount of data. The model builds on the great success of existing recurrent cells such as LSTMs: It sequences a single recurrent Memory, Attention, and Control (MAC) cell, and by careful design imposes structural constraints on the operation of each cell and the interactions between them, incorporating explicit control and soft attention mechanisms into their interfaces. We demonstrate the model's strength and robustness on the challenging CLEVR dataset for visual reasoning, achieving a new state-of-the-art 98.9% accuracy, halving the error rate of the previous best model. More importantly, we show that the new model is more computationally efficient and data-efficient, requiring an order of magnitude less time and/or data to achieve good results.

## 1    INTRODUCTION

This paper considers how best to design neural networks to perform the iterative reasoning necessary for complex problem solving. Putting facts and observations together to arrive at conclusions is a central necessary ability as we work to move neural networks beyond their current great success with sensory perception tasks (LeCun et al., 1998; Krizhevsky et al., 2012) towards displaying Artificial General Intelligence.

Concretely, we develop a novel model that we apply to the CLEVR dataset (Johnson et al., 2016) for visual question answering (VQA). VQA (Antol et al., 2015; Gupta, 2017) is a challenging multimodal task that requires responding to natural language questions about images. However, Agrawal et al. (2016) show how the first generation of successful models on VQA tasks tend to acquire only superficial comprehension of both the image and the question, exploiting dataset biases rather than capturing a sound perception and reasoning process that would lead to the correct answer (Sturm, 2014). CLEVR was created to address this problem. As illustrated in figure 1, instances in the dataset consist of rendered images featuring 3D objects of several shapes, colors, materials and sizes, coupled with unbiased, compositional questions that require an array of challenging reasoning skills such as following transitive relations, counting objects and comparing their properties, without allowing any shortcuts around such reasoning. Notably, each instance in CLEVR is also accompanied by a tree-structured functional program that was both used to construct the question and reflects its reasoning procedure – a series of predefined operations – that can be composed together to answer it.

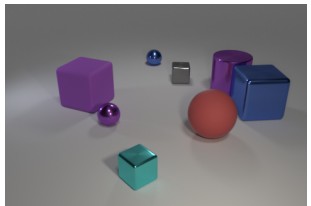

Figure 1: A sample image from the CLEVR dataset, with a question: "There is a purple cube behind a metal object left to a large ball; what material is it?"

Most neural networks are essentially very large correlation engines that will hone in on any statistical, potentially spurious pattern that allows them to model the observed data more accurately. In contrast, we seek to create a model structure that requires combining sound inference steps to solve a problem instance. At the other extreme, some approaches adopt symbolic structures that resemble the expression trees of programming languages to perform reasoning (Andreas et al., 2016b; Hu et al., 2017). In particular, some approaches to CLEVR use the supplied functional programs for supervised or semi-supervised training (Andreas et al., 2016a; Johnson et al., 2017). Not only do we wish to avoid using such supervision in our work, but we in general suspect that the rigidity of these structures and the use of an inventory of operation-specific neural modules undermines robustness and generalization, and at any rate requires more complex reinforcement learning methods.

To address these weaknesses, while still seeking to use a sound and transparent underlying reasoning process, we propose Compositional Attention Networks, a novel, fully differentiable, non-modular architecture for reasoning tasks. Our model is a straightforward recurrent neural network with attention; the novelty lies in the use of a new Memory, Attention and Composition (MAC) cell. The constrained and deliberate design of the MAC cell was developed as a kind of strong structural prior that encourages the network to solve problems by stringing together a sequence of transparent reasoning steps. MAC cells are versatile but constrained neural units. They explicitly separate out memory from control, both represented recurrently. The unit contains three sub-units: The control unit updates the control representation based on outside instructions (for VQA, the question), learning to successively attend to different parts of the instructions; the read unit gets information out of a knowledge base (for VQA, the image) based on the control signal and the previous memory; the write unit updates the memory based on soft self-attention to previous memories, controlled by the retrieved information and the control signal. A universal MAC unit with a single set of parameters is used throughout the reasoning process, but its behavior can vary widely based on the context in which it is applied – the input to the control unit and the contents of the knowledge base. With attention, our MAC network has the capacity to represent arbitrarily complex acyclic reasoning graphs in a soft manner, while having physically sequential structure. The result is a continuous counterpart to module networks that can be trained end-to-end simply by backpropagation.

We test the behavior of our new network on CLEVR and its associated datasets. On the primary CLEVR reasoning task, we achieve an accuracy of 98.9%, halving the error rate compared to the previous state-of-the-art FiLM model (Perez et al., 2017). In particular, we show that our architecture yields better performance on questions involving counting and aggregation. In supplementary studies, we show that the MAC network learns more quickly (both in terms of number of training epochs and training time) and more effectively from limited amounts of training data. Moreover, it also achieves a new state-of-the-art performance of 82.5% on the more varied and difficult human-authored questions of the CLEVR-Humans dataset. The careful design of our cell encourages compositionality, versatility and transparency. We achieve these properties by defining attention-based interfaces that constrict the cell's input and output spaces, and so constrain the interactions both between and inside cells in order to guide them towards simple reasoning behaviors. Although each cell's functionality has only a limited range of possible continuous reasoning behaviors, when chained together in a MAC network, the whole system becomes expressive and powerful. In the future, we believe that the architecture will also prove beneficial for other multi-step reasoning and inference tasks, for instance in machine comprehension and textual question answering.

## 2   RELATED WORK

There have been several prominent models that address the CLEVR task. By and large they can be partitioned into two groups: module networks, which in practice have all used the strong supervision provided in the form of tree-structured functional programs that accompany each data instance, and large, relatively unstructured end-to-end differentiable networks that complement a fairly standard stack of CNNs with components that aid in performing reasoning tasks. In contrast to modular approaches (Andreas et al., 2016a;b; Hu et al., 2017; Johnson et al., 2017), our model does not require additional supervision and makes use of a single computational cell chained in sequence (like an LSTM) rather than a collection of custom modules deployed in a rigid tree structure. In contrast to augmented CNN approaches (Santoro et al., 2017; Perez et al., 2017), we suggest that our approach provides an ability for relational reasoning with better generalization capacity and higher

computational efficiency. These approaches and other related work are discussed and contrasted in more detail in the supplementary material in section C.

# 3    Compositional Attention Networks

Compositional Attention Networks is an end-to-end architecture for question-answering tasks that sequentially performs an explicit reasoning process by stringing together small building blocks, called MAC cells, each is responsible for performing one reasoning step.

We now provide an overview of the model, and a detailed discussion of the MAC cell. The model is composed of three components: an Input unit, the core MAC network, and an output unit. A TensorFlow implementation of the network, along with pretrained models will be made publicly available.

In this paper we explore the model in the context of VQA. However, it should be noted that while the input and output units are naturally domain-specific and should be designed to fit the task at hand, the MAC network has been designed to be generic and more broadly applicable, and may prove useful in contexts beyond those explored in the paper, such as machine comprehension or question answering over knowledge bases, which in our belief is a promising avenue for future work.

## 3.1    The Input Unit

The input unit processes the raw inputs given to the system into distributed vector representations. It receives a text question (or in general, a *query*), and an image (or in general, a *Knowledge Base* (KB)) and processes each of them with a matching sub-unit, for the query and the KB, here a biLSTM and a CNN. More details can be found in the supplementary material, section A.

At the end of this stage, we get from the query sub-unit a series of biLSTM output states, which we refer to as *contextual words*, $[cw_1, ..., cw_S]$, where $S$ is the length of the question. In addition, we get $q = [\overleftarrow{cw_1}, \overrightarrow{cw_S}]$, the concatenation of the hidden states from the backward and forward LSTM passes. We refer to $q$ as the *question representation*. Furthermore, we get from the Knowledge-Base sub-unit a static representation of the knowledge base. For the case of VQA, it will be represented by a continuous matrix $KB_V$ of dimension $H, W, d$, where $H = W = 14$ are the height and width of the transformed image, corresponding to each of its regions.

## 3.2    The MAC cell

The MAC network, which is the heart of our model, chains a sequence of small building blocks, called MAC cells, each responsible for performing one reasoning step. The model is provided access to a Knowledge Base (KB), which is, for the specific case of VQA, the given image, and then upon receiving a query, i.e. a question, the model iteratively focuses, in $p$ steps, on the query's various parts, each reflects in turn the current reasoning step, which we term the *control*. Consequently, guided by this control, it retrieves the relevant information from the KB, that is then passed to the next cell in a recurrent fashion.

Drawing inspiration from the Model-View-Controller paradigm used in software design and from the commonly exercised separation between control and data paths in computer architecture, the MAC cell is composed of three units: control unit, read unit and write unit. Each has a clearly defined role and an interface through which it interacts with the other units. See figure 2.

The careful design and imposed interfaces that constrain the interaction between the units inside the MAC cell, as described below, serve as structural prior that limits the space of hypotheses it can learn, thereby guiding it towards acquiring the intended reasoning behaviors. As such, this prior facilitates the learning process and mitigate overfitting issues.

In particular, and similar in spirit to Perez et al. (2017), we allow the question to interact with the Knowledge Base – the image for the case of VQA, only through indirect means: by guiding the cell to attend to different elements in the KB, as well as controlling its operation through gating mechanisms. Thus, in both cases, the interaction between these mediums, visual and textual, or knowledge and query, is mediated through probability distributions, either in the form of attention maps, or as gates, further detailed below. This stands in stark contrast to many common approaches that fuse the

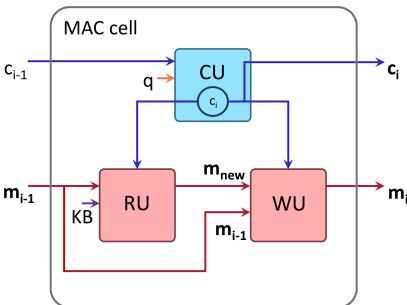

Figure 2: The MAC cell, which is a recurrent unit comprised of a Control Unit, Read Unit, and Write Unit. Blue shows the control flow and red shows the memory flow. See section 3.2 for details.

question and image together into the same vector space through linear combinations, multiplication, or concatenation. Rather, our controlled interaction distills the influence that the query should have in processing the Knowledge Base, casting it onto discrete probability distributions instead.

The MAC cell has been designed to replace the discrete and predefined "modules" used in the modular approach (Andreas et al., 2016a;b; Hu et al., 2017; Johnson et al., 2017). Rather, we create one universal and versatile cell that is applied across all the reasoning steps, sharing both its architecture as well as its parameters, across all of its instantiations. In contrast to the discrete modules, each trained to specialize to some specific elementary reasoning task, the MAC cell is capable of demonstrating a continuous range of possible reasoning behaviors conditioned on the context in which it is applied – namely, the inputs it receives from the prior cell.

Each cell $MAC_i$ maintains two dual states: *control* $c_i$ and *memory* $m_i$, both are continuous vectors of dimension $d$. The control $c_i$ represents the reasoning operation the MAC cell should accomplish in the current step – focusing only on some aspect of the whole question. This is represented by a weighted-average attention-based sum of the question words. The memory $m_i$ represents the current context information deemed relevant to respond to the query, or answer the question.This is represented practically by a weighted average over elements from the KB, or for the case of VQA, regions in the image. $m_0$ and $c_0$ are initialized each to a random vector parameter of dimension $d$. The memory and control states are passed from one cell to the next in a recurrent fashion, and used in a way reminiscent of Key-Value memory networks (Miller et al., 2016), as discussed below.

### 3.2.1 THE CONTROL UNIT

The control unit determines the reasoning operation that should be applied at this step. It receives the contextual words $[cw_1, ..., cw_S]$, the question representation $q$, and the control state from the previous MAC cell $c_{i-1}$, all of which are vectors of dimension $d$.

We would like to allow our MAC cell to perform continuously varied and adaptive range of behaviors, as demanded by the question. Therefore, we define the behavior of each cell to be a function of the contextual words $[cw_1, ..., cw_S]$, weighted-averaged according to the attention distribution that the control unit produces at each step. This will allow the cell to adapt its behavior – the reasoning operation it performs – to the question it receives, instead of having a fixed set of predefined behaviours as is the case in competing approaches Andreas et al. (2016a;b); Johnson et al. (2017).

The formal specification of the control unit is shown in figure 3. The question $q$ is linearly transformed into a vector $q_i$ of the same dimension, which in turn is concatenated with the previous control state $c_{i-1}$ and linearly transformed again to a $d$-dimensional vector $cq_i$.

$$q_i = W_i^{d,d} \cdot q + b_i^d \tag{1}$$

$$cq_i = W^{2d,d} [q_i, c_{i-1}] + b^d \tag{2}$$

Note that in contrast to all other parameters of the cell, which are shared across its instantiations at the different steps $i = 1, ..., p$, the parameters $W_i^{d,d}$ and $b_i^d$ are different for each iteration. This

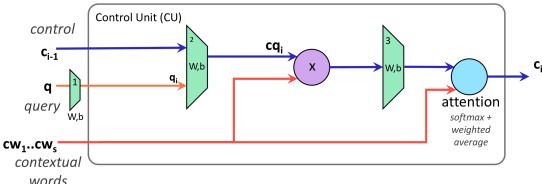

Figure 3: The Control Unit (CU) of the MAC cell. See section 3.2.1 for details. Best viewed in color.

is done to allow each cell to attend more readily to different aspects (i.e. parts) of the questions, depending on the index of the current step – its relative stage in the context of the whole reasoning process.

$cq_i$ represents the current reasoning operation we would like to perform in a continuous way, taking into account both the overall meaning of the question $q_i$, as well as the words the model attended to in the previous step, $c_{i-1}$.

However, we would like to prevent the cell from diverging in the reasoning operations it tries to perform, and instead anchor it back in the question words, by using them to represent the reasoning operation of the current step. We can achieve that by computing an attention distribution $cv_i$ over the contextual words $[cw_1, ..., cw_S]$ based on their similarity to $cq_i$. Then, summing the contextual words according to the attention distribution $cv_i$ will allow us to have a new control state, $c_i$, which is represented again in terms of words from the question. Intuitively, it is the gist of the question that is relevant to the reasoning operation we would like to perform in the current step.

$$cv_{i,s} = \text{softmax}(W^{d,1}(cq_i \circ cw_s) + b^1) \tag{3a}$$

$$c_i = \sum_{s=1}^{S} cv_{i,s} \cdot cw_s \tag{3b}$$

Finally, the control unit returns the current control state $c_i$, along with an attention map $cv_i$ over the contextual words.

### 3.2.2 THE READ UNIT

The Read Unit is provided with access to the knowledge base $KB_V$, along with the previous memory state $m_{i-1}$ and the current control $c_i$. It is responsible for retrieving relevant content from the Knowledge Base $KB_V$ for the reasoning task that the MAC cell should accomplish at this step, which is represented by the current control state $c_i$, as explained above. Figure 4 shows a diagram.

The relevance of the new information is judged in two stages by the "relatedness" of each element in the KB (or for the case of VQA, each region in the image) to either the memory $m_{i-1}$ that has accumulated relevant information from previous iterations, or to the current control $c_i$, pointing towards the next piece of information that should be taken into account. Here, relatedness is measured by trained linear transformations comparing each element to the previous memory and the current control.

More formally, at the first stage, the interaction between each element $KB_{h,w}$, where $h = 1, ..., H, w = 1, ..., W$, and the previous memory $m_{i-1}$ is computed by:

$$m'_{i-1} = W^{d,d} \cdot m_{i-1} + b^d \tag{4}$$

$$KB'_{h,w} = W^{d,d} \cdot KB_{h,w} + b^d \tag{5a}$$

$$(I_{m-KB})_{h,w} = m'_{i-1} \circ KB'_{h,w} \tag{5b}$$

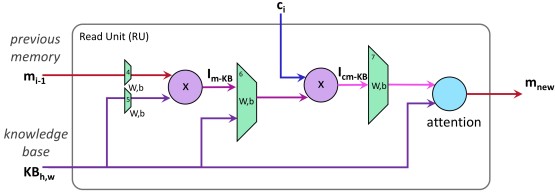

Figure 4: The Read Unit (RU) diagram. Blue refers to control flow, purple to knowledge flow and red to memory flow. See section 3.2.2 for description.

These memory-KB interactions measure the relatedness of each element in the KB to the memory accumulated so far, which holds information that has been deemed relevant to handle previous reasoning steps towards addressing the question. They allow the model to perform transitive inference, retrieving a new piece of information that now seems important in light of the recent memory retrieved in a prior iteration.

However, there are cases which necessitate the model to temporarily ignore current memories, when choosing the new information to retrieve. Logical OR is a classical example: when the model has to look at two different objects at the same time, and assuming it stored one of them at the first iteration, it should briefly *ignore* it, considering *new information that is relevant to the question but is unrelated to the memory*. In order to achieve such capability, the read unit concatenates the original KB elements to each corresponding memory-KB interaction, which are then projected back to $d$-dimensional space (equation 6a):

$$I_{m-KB}{}' = W^{2d,d} \left[ I_{m-KB}, KB_{h,w} \right] + b^d \tag{6a}$$

$$I_{cm-KB} = c_i \circ \left( I_{m-KB} \right)' \tag{6b}$$

At the second stage, the read unit compares the current $c_i$ with these memory-KB interactions, in order to focus on the information that is relevant to the current reasoning operation that the MAC cell seeks to accomplish. The result is then passed to a softmax layer yielding an attention map $mv_i$ over the KB, which is used in turn to retrieve the relevant information to perform the current reasoning step.

$$mv_i = \text{softmax} \left( W^{d,d} \cdot I_{cm-KB} + b^d \right) \tag{7a}$$

$$m_{new} = \sum_{h,w=1,1}^{H,W} (mv_i)_{h,w} \cdot KB_{h,w} \tag{7b}$$

Finally, the read unit returns the newly retrieved information $m_{new}$, along with an attention map $mv_i$ over the Knowledge Base.

To give an example of the read unit operation, assume a given question $q$ such as "What object is located left to the blue ball?", whose associated answer is "cube". Initially, no cue is provided to the model to attend to that cube, since no direct information about it presents in the question. Instead, based on its comprehension of the question, the model may start by focusing on the blue ball at the first iteration, such that the memory state $m_1$ will capture the blue ball. However, in the second iteration, the control unit, after re-examining the question, may realize it should now look left, storing the word "left" in $c_2$. Then, when considering *both* $m_1$ and $c_2$, the read unit will realize it should perform a reasoning operation corresponding to the word "left" (stored in $c_2$) given a memory representing the blue ball in $m_1$, thereby allowing it to look left to the blue ball and find the cube.

### 3.2.3 THE WRITE UNIT

The Write Unit is responsible for creating the new memory state $m_i$ that will reflect all the information considered to be important to answer the question so far, i.e. up to the current iteration in the

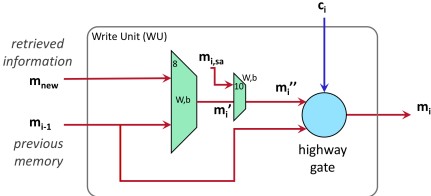

Figure 5: The Write Unit (WU) diagram. Blue refers to control flow and red to memory flow. See section 3.2.3 for description.

reasoning process. It receives the last memory state $m_{i-1}$ from the previous MAC cell, along with the newly retrieved information from the read unit in the current iteration, $m_{new}$. See figure 5 for a diagram.

In the main design we have explored, merging the new information with the previous memory state is done simply by a linear transformation.

$$m_i' = W^{2d,d}[m_{new}, m_{i-1}] + b^d \tag{8}$$

In addition, we have explored two variations of this design. The first, self-attention, allows considering any previous memories rather than just the last one $m_{i-1}$, thus providing the network with the capacity to model non-sequential reasoning processes. The second variation is adding gating mechanisms to the writing unit. These may allow the model to dynamically adjust the practical length of the computation to the question complexity and stabilize the memory content throughout the sequential network (similarly to GRUs and LSTMs).

**Self-Attention.**   The current architecture that we have presented allows the model to perform reasoning steps in a sequence, passing control and memory states from one cell to the following. However, we would like to grant the system with more flexibility. Particularly, we would like to allow it to capture more complicated reasoning processes such as trees and graphs - Directed Acyclic Graph (DAG) in particular, where several branches of reasoning sub-processes are merged together in later stages. Indeed, the CLEVR dataset includes cases where the questions embody tree-like reasoning process, rather than just sequences, which we would like to address correctly in our model.

We achieve that by adding self-attention connections between each MAC cell and all the prior cells. Since each cell can look on all the prior reasoning steps and their corresponding memories retrieved from the Knowledge Base, it can virtually capture any directed acyclic graph, while still having physically sequential layout.

More formally, the current MAC cell, of the $i^{th}$ iteration, is granted with access to $c_1, ..., c_{i-1}$ along with the corresponding $m_1, ..., m_{i-1}$, that have been computed by the prior MAC cells. It begins by computing the similarity between $c_i$ and $c_1, ..., c_{i-1}$, and use it to derive an attention map over the prior MAC cells $sa_{i,j}$ for $j = 1, ..., i-1$. This represents the relevance of the $j^{th}$ prior reasoning step to the current one $i$ (equation 9a).

Then, we average the previous *memories* according to this resulted attention map $sa_{ij}$. We obtain $m_{sa}$, representing the information from all the other reasoning steps that is relevant to the current one (equation 9b).

This resembles the approach of Key-Value networks (Miller et al., 2016). The similarity between control states, corresponding to the reasoning operations that are performed in each prior step, allows the model to select which memories should be taken into account, when creating the new memory – namely, which branches of the reasoning process should be merged together at this point.

$$sa_{ij} = \text{softmax}\left(W^{d,1}(c_i \circ c_j) + b^1\right) \tag{9a}$$

$$(m_{sa})_i = \sum_{j=1}^{i-1} sa_{ij} \cdot m_j \tag{9b}$$

Finally, we use $m_{sa}$ along with $m_i'$ to compute $m_i''$, the new memory content in this variation.

$$m_i'' = W^{2d,d}[m_{new}, m_i'] + b^d \tag{10}$$

**Memory Gate.** The currently presented MAC network has some fixed number $p$ of concatenated MAC cells, representing the length of the overall reasoning process we perform. However, not all questions require reasoning sequence of the same length. Some questions are simpler while others more complex.

Motivated by this observation, we add a gate over the new memory computed at each step, that may selectively keep content of the previous memory $m_{i-1}$ unchanged. Practically, the gate functions in a similar way to a highway network (Srivastava et al., 2015), where the gate value is conditioned on the current reasoning operation, $c_i$.

$$c_i' = W^{d,d} \cdot c_i + b^d \tag{11a}$$
$$m_i = \text{sigmoid}\,(c_i') \cdot m_{i-1} + (1 - \text{sigmoid}\,(c_i')) \cdot m_i'' \tag{11b}$$

The write unit returns the new memory state $m_i$, that will be passed along with $c_i$ to the next MAC cell.

### 3.2.4 DISCUSSION

Overall, when designing the MAC cell, we have attempted to formulate the inner workings of an elementary, yet generic reasoning skills: the model decomposes the problem into steps, focusing on one at a time. At each such step, it takes into account:

- **The control** $c_i$: Some aspect of the task - pointing to the **future** work that has left to be done.

- **The previous memory or memories**: The partial solution or evidence the cell has acquired so far – pointing to the **past** work that has already been achieved.

- **The newly retrieved information** $m_{new}$: that is retrieved from the knowledge base $KB$ and may or may not be transitively related to that partial solution or evidence - the **present**, or current work.

Considering these three sources of information together, the cell finally adds the new information up into its working memory, $m_i$, progressing one more step towards the final answer.

### 3.3 THE OUTPUT UNIT

The output unit receives the question representation $q$, along with the memory state passed from the last MAC cell $m_p$, where $p$ is the number of MAC cells in the network – representing the number of reasoning steps in the whole process. It inspects both and predicts an answer based on their concatenation. Intuitively, we would like our model to consider both the question as well as the relevant information that has been progressively retrieved from the KB, deemed the necessary information to answer it.

Note that considering both $q$ and $m_p$ is critical to answer the question. While $m_p$ represents the information collected from KB, we still need to recall what has been asked about it to be able to answer accordingly. This is especially true in our case, when all other interactions between the question and the KB are mediated through attention distributions, rather than being transformed into a shared continuous vector space.

The prediction is built out of a standard 2-layers fully-connected softmax-based classifier with hidden dimension $d$ and output dimension that matches the number of possible answers in the dataset. The classifier receives $[m_p, q]$ as input and returns a probability distribution over the answers.

Table 1: CLEVR accuracy by baseline methods, competing methods, and our method (MAC) - for our method we show test results with GLoVE initialization and validation results with and without GLoVE. (*) denotes use of extra supervisory information through program labels. ($\dagger$) denotes use of data augmentation. ($\ddagger$) denotes training from raw pixels. ($\diamond$) denotes GLoVE word vector initialization.

| Model | Overall | Count | Exist | Compare Numbers | Query Attribute | Compare Attribute |
|---|---|---|---|---|---|---|
| Human (Johnson et al., 2017) | 92.6 | 86.7 | 96.6 | 86.5 | 95.0 | 96.0 |
| Q-type baseline (Johnson et al., 2017) | 41.8 | 34.6 | 50.2 | 51.0 | 36.0 | 51.3 |
| LSTM (Johnson et al., 2017) | 46.8 | 41.7 | 61.1 | 69.8 | 36.8 | 51.8 |
| CNN+LSTM (Johnson et al., 2017) | 52.3 | 43.7 | 65.2 | 67.1 | 49.3 | 53.0 |
| CNN+LSTM+SA (Johnson et al., 2016) | 76.6 | 64.4 | 82.7 | 77.4 | 82.6 | 75.4 |
| N2NMN* (Hu et al. 2017) | 83.7 | 68.5 | 85.7 | 84.9 | 90.0 | 88.7 |
| PG+EE (9K prog.)* (Johnson et al., 2017) | 88.6 | 79.7 | 89.7 | 79.1 | 92.6 | 96.0 |
| PG+EE (700K prog.)* (Johnson et al., 2017) | 96.9 | 92.7 | 97.1 | 98.7 | 98.1 | 98.9 |
| CNN+LSTM+RN$^{\dagger\ddagger}$ (Santoro et al., 2017) | 95.5 | 90.1 | 97.8 | 93.6 | 97.9 | 97.1 |
| CNN+GRU+FiLM (Perez et al., 2017) | 97.7 | 94.3 | 99.1 | 96.8 | 99.1 | 99.1 |
| CNN+GRU+FiLM$^{\ddagger}$ (Perez et al., 2017) | 97.6 | 94.3 | 99.3 | 93.4 | **99.3** | 99.3 |
| CNN+GRU+FiLM (Perez et al., 2017) | 97.7 | 94.3 | 99.1 | 96.8 | 99.1 | 99.1 |
| MAC (this paper)-val | 98.9 | 97.1 | 99.5 | 99.3 | 99.2 | 99.2 |
| MAC-val$^{\diamond}$ | 99.0 | 97.2 | 99.5 | 99.5 | 99.5 | 99.4 |
| **MAC-test$^{\diamond}$** | **98.9** | **97.2** | **99.5** | **99.4** | **99.3** | **99.5** |

# 4 EXPERIMENTS

We evaluate our model on the recent CLEVR dataset (Johnson et al., 2016). CLEVR is a synthetic dataset consisting of 700K tuples; each consists of a 3D-rendered image featuring objects of various shapes, colors, materials and sizes, coupled with compositional multi-step questions that measure performance on an array of challenging reasoning skills such as following transitive relations, counting objects and comparing their properties. In addition, each question is associated with a formal program, specifying the reasoning operations that should be performed to compute the answer, among 28 possibilities.

We first perform experiments on the original 700k CLEVR dataset (Johnson et al., 2016), comparing to prior work. As shown in table 1, our model matches or outperforms all existing models both in overall accuracy, as well as in each category, testing different reasoning skills. In particular, for the overall performance, we achieve 98.94% accuracy, more than halving the error rate of the prior best model, FiLM (Perez et al., 2017).

**Counting and Numerical Comparison.** Remarkably, our performance on questions testing counting and numerical comparisons is significantly higher than the competing models, which consistently struggle on this question type. Again, we nearly halve the corresponding error rate. These results demonstrate the aptitude of attention mechanisms to perform counting, reduction and aggregation, in contrast to alternative, CNN-based approaches.

**Training Length and Computational-Efficiency.** We examine the learning curves of our and competing models. We have trained all models on the same architecture and used the author code for the other models. Aiming at having equal settings for comparison, we ran all models including ours with learned random words vectors. In order to make sure the results are statistically significant we ran each model multiple (10) times, and plotted the averages and confidence intervals (figure 4). The results show that our model learns significantly faster than the other leading methods, FiLM (Perez et al., 2017) and PG+EE (Johnson et al., 2017). While we do not have learning curves for the Relational Network model, Santoro et al. (2017) report approximately 1.4 million iterations to achieve 95.5% accuracy, which are equivalent to 125 epochs approximately, whereas our model achieves a comparable accuracy after 3 epochs only, yielding 40x reduction in the length of the training process.

Naturally, the smaller number of required training steps also translates to comparably shorter training time. Perez et al. (2017) report training time of 4 days, equivalent to 80 epochs, to reach accuracy of 97.7%. In contrast, we achieve higher accuracy in 6 epochs, taking 9.5 hours overall, leading to 10x reduction in training time.

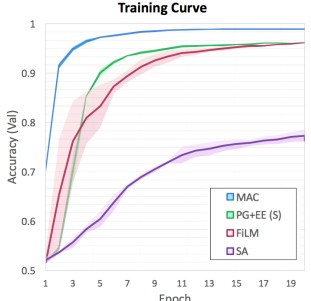 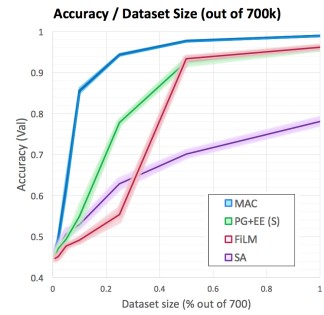

Figure 6: Training curves and accuracies for MACs (our model), FiLM (Perez et al., 2017), PG+EE (Johnson et al., 2017) and stacked-attention (Yang et al., 2016; Johnson et al., 2017). (Note: PG+EE uses the supported CLEVR programs as strong supervision.) *Left:* Training curve (accuracy/epoch). *Right:* Learning curve: Accuracy for 1%, 2%, 5%, 10%, 25%, 50% and 100% of the 700k CLEVR samples.

Table 2: Accuracy on CLEVR-Humans of previous methods and our method (MAC), before (left) and after (right) fine-tuning on the CLEVR-Humans training data. PG+EE uses supervised data. For our method (MAC) we show test results with GLoVE initialization and validation results with and without GLoVE.

| Model | Train CLEVR | Train CLEVR + fine-tune HUMANS |
|---|---|---|
| LSTM (Johnson et al., 2017) | 27.5 | 36.5 |
| CNN+LSTM (Johnson et al., 2017) | 37.7 | 43.2 |
| CNN+LSTM+SA+MLP (Johnson et al., 2016) | 50.4 | 57.6 |
| PG+EE (18K prog.)* (Johnson et al., 2017) | 54.0 | 66.6 |
| CNN+GRU+FiLM (Perez et al., 2017) | 56.6 | 75.9 |
| MAC-val | 56.9 | 81.2 |
| MAC-val$^\diamond$ | 58.5 | 82.0 |
| **MAC-test$^\diamond$** | **58.6** | **82.5** |

## 4.1 DATA EFFICIENCY

We have explored the performance of our and other leading approaches on smaller subsets of the CLEVR dataset, in order to study the ability of models to generalize from smaller amount of data. We sampled at random subsets of CLEVR, with 10%, 25% and 50% of its original 700k size, and used them to train our and other 3 proposed models for the CLEVR task: FiLM (Perez et al., 2017), the strongly-supervised PG+EE (Johnson et al., 2017), and stacked-attention networks (Johnson et al., 2017; Yang et al., 2016).

As shown in figure 4, our model outperforms the other models by a wide margin for all subsets of the CLEVR dataset. For 50% of the data, equivalent to 350k samples, other models obtain accuracies ranging between 70% and 92%, while our model achieves 97.9%. The gap becomes larger as the dataset size reduces: for 25% of the data, equivalent to 175k samples, performance of other models is between 50% and 77%, while our model maintains a high 95.4% accuracy.

Finally, for 10% of the data – 70k samples, still a sizeable amount – our model is the only one that manages to generalize, with performance of 84.7% on average, whereas the other three models fail, achieving 47.6%-57.5% . Note that as pointed out by (Johnson et al., 2016) a simple baseline that predicts the most frequent answer for each of the question types achieves already 42.1%, suggesting that answering half of the questions correctly means that the competing models barely learn to generalize from the smaller dataset. These results demonstrate the robustness of our architecture and its key role as a structural prior guiding our network to learn the intended reasoning skills.

## 4.2 CLEVR HUMANS - NATURAL LANGUAGE QUESTIONS

We analyze our model performance on the CLEVR-Humans dataset (Johnson et al., 2017), consisting of natural language questions collected through crowdsourcing. As such, the dataset has diverse vocabulary and linguistic variations, and it also demands more varied reasoning skills.

Since the training set is relatively small, consisting of 18k samples, we use it to finetune a model pretrained on the standard CLEVR dataset. However, since most of the vocabulary in CLEVR-Humans is not covered by CLEVR, we do not train the word vectors during the pre-training stage, so to prevent drift in their meaning compared to other uncovered words in CLEVR-Humans that may be semantically related.

As shown in table 2, our model achieves state-of-the-art performance on CLEVR-Humans both before and after fine-tuning. It surpasses the next-best FiLM model, (Perez et al., 2017) by 6.6% percent, achieving 82.5%.

The results substantiate the model's robustness against linguistic variations and noise, as well as its ability to adapt to diverse vocabulary and varied reasoning skills. Arguably, the soft attention performed over the question words allows the model to focus on the words that are most critical to answer the question and translate them to corresponding reasoning operations, giving less attention to irrelevant linguistic variations.

## 4.3 ABLATIONS

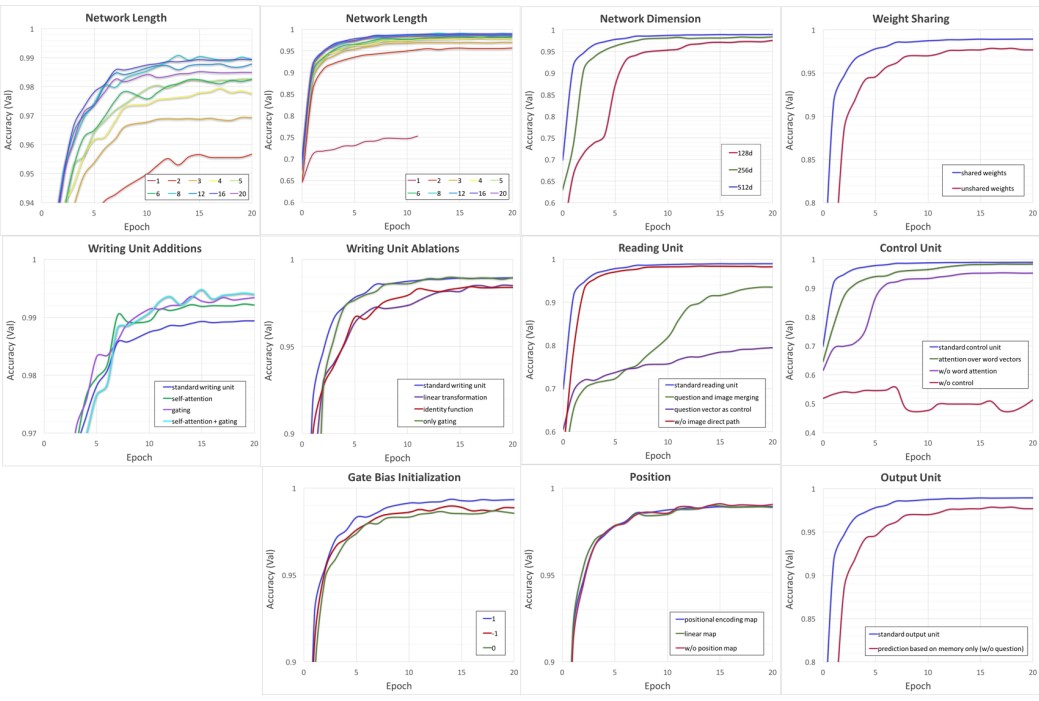

Figure 7: Ablations and variations of the MAC network. See text for full detail.

Based on the validation set, we have conducted an ablation study on our model to understand better the contribution of each of its component to the overall performance. We tested each setting on the standard 700K CLEVR dataset as well as on 10% subset of the dataset. See table 3 for the numerical results. In addition, figure 4.3 presents the training curves for the different settings trained on the standard dataset. Overall, the results demonstrate the robustness of the model to hyperparameter variations such as network dimension and length, and also the impact of different aspect and components of MAC on its performance.

Table 3: Ablations Results for the full standard CLEVR dataset and 10% subset of it. See text for full details.

| Model | Standard CLEVR | 10% CLEVR |
|---|---|---|
| MAC standard model (dimension 512) | 98.9 | 84.5 |
| dimension 256 | 98.4 | 76.3 |
| dimension 128 | 97.6 | 77.0 |
| w/o control | 55.6 | 51.5 |
| w/o word-attention | 95.3 | 63.2 |
| full-question-as-control | 80.7 | 65.0 |
| w/o positional encoding | 99.1 | 80.0 |
| linear position encoding | 99.0 | 83.2 |
| identity writing unit | 98.5 | 84.5 |
| only-gating writing unit | 99.3 | 83.1 |
| + self-attention | 83.2 | 99.2 |
| + gating | 99.4 | 83.1 |
| + self-attention and gating | 99.5 | 85.5 |
| gate bias 0 | 98.7 | 84.9 |
| gate bias 1 | 99.4 | 68.5 |
| gate bias -1 | 99.0 | 77.1 |
| unshared weights | 97.8 | 67.5 |
| prediction without question information | 97.8 | 64.7 |

**Network Length.** We have tested the model performance as a function of the network's length – the number of MAC cells that were sequenced together. The results show the positive correlation between the network length and its performance. We can see that for 1 cell the scores are relatively low – 75%, but adding at least one more cell leads to a significant increase in performance above 95%. The performance keeps improving up to lengths 8-16 that achieve 98.9-99.1%. The results also teach us about the complexity of the dataset, by showing the relatively significant benefits of having at least 4 cells, each modeling a reasoning step.

**Network Dimension.** We have varied the state dimension to check the robustness of the model to hyperparameters. The results on the standard CLEVR dataset show the model is able to maintain high performance with dimension of 128, albeit after a longer training process, achieving 97.6%, compared to 98.94% achieved with dimension of 512. However, for 10% of CLEVR, the larger 512-dimension allows accuracy increase by 7.5% over dimension of 128.

**Weight Sharing.** We have tested the impact of sharing weights between cell has on the model performance for network of length $p = 12$. The results show that for the standard dataset there is only a small difference between these settings of 1%. However, for less data, we see much more significant drop 16.9% in the unshared-parameters setting compared to the shared one. Indeed, we observe that a model with less parameter is more data-efficient and has a lower tendency to overfit the data.

**Control Unit.** We have performed several ablations in the control unit to understand its contribution to the overall model performance. Based on the results, first, we can see the the question information is crucial for the model to handle the questions, as can be noted by the low performance of the model when there is no use of control signal whatsoever. Second, we have tested the model performance when using the continuous control state computed by question (2) in section 3.2.1, without having word-attention, in order to understand its relative contribution. Based on the results, we can indeed see that using word-attention is useful for accelerating the training process and achieving higher accuracies both for the standard dataset as well as for the small subset, where using word-attention increases results in 21.4%. We also see that using the "contextual words" produced by the question-unit LSTM is useful in accelerating the model performance, when compared to using the word-vectors directly.

**Reading Unit.** We have conducted several ablations for the reading unit to better understand its behavior and contribution to the performance of the model. The standard MAC reading unit uses the control state – which averages the question words based on attention distributions computed per each reasoning step. In this ablation experiment, we have tested using the full question representation $q$ instead across all reasoning steps to gain better understanding of the the contribution of word-attention to the model performance. Indeed, we can see that using $q$ rather then the control state $c_i$ results in a significant drops in performance – 19.4% for the full CLEVR dataset and 19.5% for 10% of the data.

We have conducted additional ablation experiment to better understand the contribution of using the KB features directly in the first-stage information retrieval process described in section 3.2.2, compared to using only the dot-products of the KB elements with the previous memory state $m_{i-1}$. For the full CLEVR dataset, we can see that this component has only a small impact in the final performance - ultimately resulting in 0.06% performance difference. However, for the 10% of the data, we can see that the difference in performance when ablating this component is much larger - 11.2%.

**Writing Unit Ablations.** In our main MAC model variant, the memory unit merges the new information $m_{new}$ with the previous memory state $m_{i-1}$ by combining them through a linear transformation. In this experiment, we have explored other variations, such as assigning $m_{new}$ to $m_i$ directly – ignoring previous memories, or doing a linear transformation based on $m_{new}$ only. The results show that in fact such variant is only slightly worse than our main variant – 0.4%. We also conducted an experiment in which we merge the new information with the previous memory just by a having a gate that does a weighted average of them. The results show that this variant performs equivalently to our standard linear-transformation variant.

**Writing Unit Additions.** We have explored the impact of the writing unit variants described in section 3.2.3 – adding self-attention, gating mechanisms, or both, compared to our standard main model that uses a linear transformation to merge the newly retrieved information $m_{new}$ with the previous memory $m_i$. For the complete CLEVR dataset we can see that indeed both these variants are very helpful in increasing the model performance. Compared to our standard MAC model that achieves 98.94% on the validation set, self-attention yields accuracy of 99.23%, gating yields 99.36% and adding both achieves 99.48%.

**Output Unit.** In our standard model, the final predictions made in the output unit are based on the final memory state $m_p$ as well as question representation $q$ (stands for the final hidden states of the backward and forwards passes of the LSTM). We have explored the contribution of basing the model prediction on the latter, by testing the model performance when prediction is based on memory alone, for the complete and 10% datasets. We can see that in both settings basing the model's predictions on the question representation allows faster training and higher accuracies. Notable is the gap in performance for the 10% CLEVR - 19.8% increase by using the question representation to make predictions. These results are very reasonable intuitively, since the model is structured such that the memory holds only information that was retrieved from the image. Thus, questions that may ask for instance on different aspects (such as color or shape) of the same object in the image may result in the same memory content, which is thus does not directly contain enough information to respond such questions.

**Position.** In our standard model, similarly to the practice of competing models (Santoro et al., 2017; Perez et al., 2017; Hu et al., 2017), we have concatenated positional information to each region of the image, in order to increase the model capability to perform spatial reasoning. We have explored both simple linear maps at a constant $[-1, 1]$ as well as more complex positional encoding suggested by (Vaswani et al., 2017). However, the results for both the standard dataset and the 10% version show a very negligible improvement at best when adding positional encoding information, demonstrating the capability of MAC to perform spatial reasoning without data augmentation.

**Gate Bias Initialization.** For our model variant with gating mechanism (described in section 3.2.3) we have tested the effect of setting different values for the gate bias - $-1, 0$ and $1$. for $-1$ the model is initialized to biased for keeping the previous memory value whereas for $1$ it will be biased for using the new memory instead. We can see that for the complete dataset setting the bias to $1$ is optimal – apparently since the model has enough data to learn to apply each cell effectively. In contrast, for the small 10% CLEVR data, setting the bias to $0$ shows better performance, biasing the model to using less cells overall which results ultimately in a theoretically-simpler model that can fit less data more effectively.

## 4.4 INTERPRETABILITY

We have looked into attention maps over the image and question that the model produces during its computation and provide a few examples in figure 4.4. The first example shows us how the model parses the question in steps, first focusing on the main entity that the question is about, then on

relation of this entity to the "brown matte thing" which is then located in the image. Finally, the model correctly focuses on the small brown cube and predicts the right answer – brown.

The second example shows a model with 4 cells instead of 6, that similarly parse the question in iterations and focuses on the relevant objects at each step, though we can see that the reasoning process looks somewhat different when the MAC network has fewer cells.

The last example shows how how the model handles counting and OR operations. It starts from identifying the task - computing a number, and then red objects as well as the cylinder, one at a time, allowing it ultimately to respond correctly, with the answer 2.

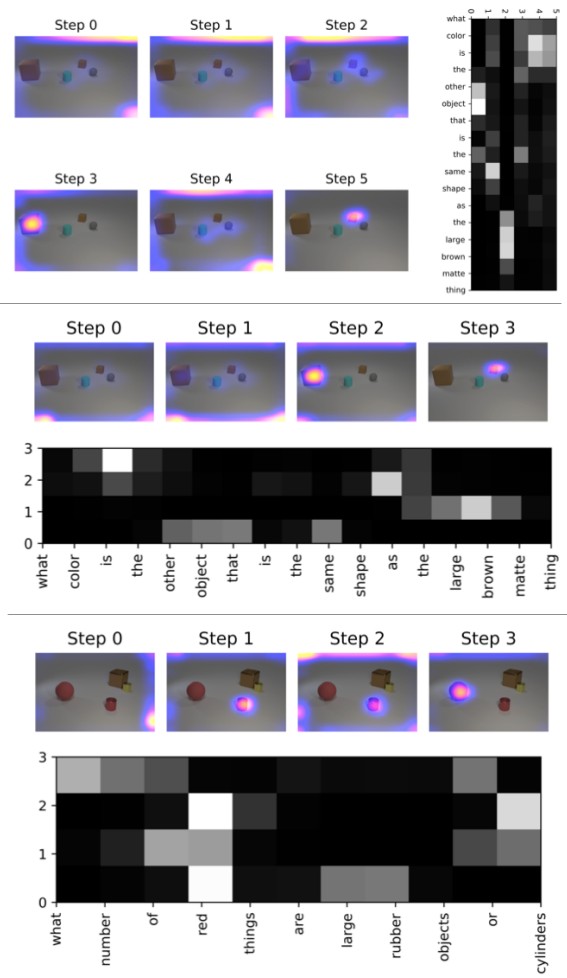

Figure 8: Three examples of attention maps for the image and the question. See text for full details.

## 5    CONCLUSION

We have given a first demonstration of how a sequence of Memory, Attention and Control (MAC) cells combined into a Compositional Attention Network provides a very effective tool for neural reasoning. In future work, we wish to explore this promising architecture for other tasks and domains, including real-world VQA, machine comprehension and textual question answering.

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

## SUPPLEMENTARY MATERIAL

## A   DETAILS OF INPUT UNIT

The input unit processes the raw inputs given to the system into distributed vector representations. It receives a text question (or in general, a *query*), and an image (or in general, a *Knowledge Base* (KB)) and processes each of them with a matching sub-unit. Here we provide details of the Query Unit and the Image Unit used in this work.

### A.0.1   THE QUERY UNIT

We encode a query of $S$ words into a continuous representation using a bidirectional LSTM (Hochreiter & Schmidhuber, 1997; Graves et al., 2013). Each word is associated with a word embedding $w_s$, where $s = 1, ..., S$. In our case, we use GloVE words embeddings (Pennington et al., 2014). Then, these embeddings are processed by a bidirectional $LSTM$ of dimension $d$ that outputs:

- a matching sequence of $d$-dimensional output states, which we refer to as *contextual words*, $[cw_1, ..., cw_S]$

- $d$-dimensional hidden state $q = [\overleftarrow{cw_1}, \overrightarrow{cw_S}]$, the concatenation of the hidden states from the backward and forward passes. We refer to $q$ as the *question representation*.

Intuitively, each contextual word $cw_s$ represents the meaning of $s^{th}$ word, in the context of the question, while the hidden state $q$ represents the overall (compositional) meaning of the question.

### A.0.2   THE IMAGE UNIT

Given an image, and following prior work on CLEVR (**?**Santoro et al., 2017; Perez et al., 2017), we extract conv4 features from ResNet101 (He et al., 2016) pretrained on ImageNet (Krizhevsky et al., 2012) which we treat as a fixed initial representation of the image, $x$ of dimension $H, W, C$ where $H = W = 14$ are the height and width of the transformed image and $C = 1024$ is the number of channels. Each feature $x_{h,w}$ represents one region in the original image.

Similar to prior work (Hu et al., 2017; Santoro et al., 2017; Perez et al., 2017), we would like to allow our model to reason explicitly about spatial locations, as required by many of the questions in CLEVR, and therefore we concatenate to this representation a spatial map that represents each of the positions in the image. However, in contrast to prior work that uses a linear meshgrid feature map with 2 features $h$ and $w$ ranging from $-1$ to 1, and to allow better representation of the positions, we use the positional encoding scheme proposed by Vaswani et al. (2017):

$$p_{(h,2i)} = \sin\left(h/10000^{2i/p_d}\right)$$

$$p_{(h,2i+1)} = \cos\left(h/10000^{2i/p_d}\right)$$

And similarly for $w$, where $p_d$ is a hyperparameter. Overall, the positional encoding of a feature at position $(h, w)$ is $[p_h, p_w]$, the concatenation of the positional encodings for $h$ and $w$.

This positional encoding scheme allows better correspondence between the distance of 2 positions $(x, y)$ and $(x, y)$ in the image and a vector similarity of their positional encodings, even when $p_d$ is larger than two.

We then concatenate the obtained spatial map with $x$, receiving a spatially-aware image representation $x_p$. Then, we pass this representation through two CNN layers with $d$ output channels and obtain a final representation of the image, which we refer to as our *Visual Knowledge Base* ($KB_V$ that is used in further components of the model.

## B    IMPLEMENTATION AND TRAINING DETAILS

For the question processing, we use GloVE (Pennington et al., 2014) word-vectors with dimension 300. For the image processing, we extract conv4 features from ResNet101 (He et al., 2016) pretrained on ImageNet (Krizhevsky et al., 2012), with dimension $H, W, C$ where $H = W = 14$ and $C = 1024$, followed by 2 CNN layers with kernel size 2. We use MAC network with $p = 12$ cells, and train it using Adam (Kingma & Ba, 2014), with learning rate $10^{-4}$. We train our model for $10 - 20$ epochs, with batch size $64$, and use early stopping based on validation accuracies. During training, the moving averages of all weights of the model are maintained with the exponential decay rate of 0.999. At test time, the moving averages instead of the raw weights are used. We use dropout $0.85$, and ELU (Clevert et al., 2015) which in our experience has reduce the training process compared to RELU. The training process takes roughly 10-20 hours on a single Titan X GPU.

## C    FURTHER DISCUSSION OF RELATED WORK

In this section we provide detailed discussion of related work. Several models have been applied to the CLEVR task. These can be partitioned into two groups, module networks that use the strong supervision provided as a tree-structured functional program associated with each instance, and end-to-end, fully differentiable networks that combine a fairly standard stack of CNNs with components that aid them in performing reasoning tasks. We also discuss the relation of MAC to other approaches, such as memory networks and neural computers.

### C.1    MODULE NETWORKS

The modular approach (Andreas et al., 2016a;b; Hu et al., 2017; Johnson et al., 2017) first translates the given question into a tree-structured action plan, aiming to imitate the ground-truth programs provided as a form of strong-supervision. Then, it constructs a tailor-made network that executes the plan on the image in multiple steps. This network is composed of discrete units selected out of a collection of predefined modules, each responsible for an elementary reasoning operation, such as identifying an objects color, filtering them for their shape, or comparing two amounts. Each module has its own set of learned parameters (Johnson et al., 2017), or even hand-crafted design (Andreas et al., 2016a) to guide it towards its intended behavior.

Overall, this approach makes discrete choices at two levels: the identity of each module – the behavior it should learn among a fixed set of possible types of behaviors, and the network layout – the way in which these modules are wired together to compute the answer progressively. Hence, their differentiability is confined to the boundaries of a single module, disallowing end-to-end training.

Several key differences exist between our approaches. First, our model replaces the fixed modules collection with one versatile and universal cell that shares both its architecture and parameters across all of its instantiations, and is applied across all the reasoning steps. Second, it replaces the dynamic recursive tree structures with a sequential topology, augmented by soft attention mechanisms, as done in Bahdanau et al. (2014). This confers our network with a virtual capacity to represent arbitrarily complex Directed Acyclic Graphs (DAGs) while still having efficient and readily deployed physical sequential structure. Together, both of these relaxations allow us to effectively train our model end-to-end by backpropagation alone, whereas module networks demand a more involved training scheme that relies on the strongly-supervised programs at the first stage, and on various Reinforcement Learning (RL) techniques at the second. Furthermore, while our model can be train without the strong supervisory programs, developing adaptive reasoning skills to address the task is it trained for, the modular approach reliance on questions structured and formal representation hinder its applicability to real-world tasks.

### C.2    AUGMENTED CONVOLUTIONAL NEURAL NETWORKS

Alternative approaches for the CLEVR task that do not rely on the provided programs as a strong supervision signal are Santoro et al. (2017) and Perez et al. (2017). Both complement standard multi-layer Convolutional Neural Networks (CNNs) with components that aid them in handling compositional and relational questions.

**Relational Networks.** Santoro et al. (2017) appends a Relation Network (RN) layer to the CNN. This layer inspects all pairs of pixels in the image, thereby enhancing the network capacity to reason over binary relations between objects. While this approach is very simple and elegant conceptually, it suffers from quadratic computational complexity, in contrast to our and other leading approaches. But beyond that, closer inspection reveals that this direct pairwise comparison might be unnecessary. Based on the analogy suggested by Santoro et al. (2017), according to which pixels are equivalent to objects and their pairwise interactions to relations, a RN layer attempts to grasp the induced graph between objects all at once in one shallow and broad layer. Conversely, our attention-based model proceeds in steps. It basically compares the image to its current memory and control for this step, aggregates the attended regions into the new memory, and repeats the process. By the same analogy, it traverses a narrow and deep path, progressively following transitive relations. Consequently, our model exhibits a relational capacity while circumventing the computational inefficiency.

**FiLM.** FiLM (Perez et al., 2017) is a recently proposed method that interleaves standard CNN layers that process the given image with linear layers, reminiscent of layer normalization techniques (Ba et al., 2016; Ioffe & Szegedy, 2015). Each of these layers, called FiLM, is conditioned on the question: the question words are processed by a GRU, and its output is linearly transformed into matching biases and variances for each of the CNN layers, tilting its activations to reflect the specifics of the given question and affect the computation done over the image.

Similarly to our model, this approach features distant modulation between the question and the image, where rather than being fused together into the same vector space, the question can affect the image processing only through constrained means – for the case of FiLM – linear transformations. However, since the same transformation is applied to all the activations homogeneously, agnostic to both their spatial location as well as the features values, this approach does not allow the question to differentiate between regions in the image based on the objects or concepts they represent – on the content of the image. This stands in stark contrast to our attention-based model, which readily allows and actually encourages the question to inform the model about relevant regions to focus on. We speculate that this still distant, yet more direct interaction between the question and the data, or image, for the case of VQA, facilitates learning and increases generalizability. It may be more suitable to VQA tasks, and CLEVR in particular, where the questions demand the responder to focus on specific objects, and reason about their properties or relations, rather than respond based only on a holistic view of the image that may lead to sub-optimal results (Yang et al., 2016), as is the case of FiLM. Indeed, as demonstrated in 4, there is significant evidence showing our models better generalization capacity, allowing it to achieve high accuracies much faster, and from less data than FiLM and other competing methods.

## C.3  MEMORY AND ATTENTION

Our architecture draws inspiration from recent research on memory and attention (Kumar et al., 2016; Xiong et al., 2016; Graves et al., 2014; 2016). Kumar et al. (2016); Xiong et al. (2016) propose the Dynamic Memory Network model that proceeds in an iterative process, applying soft attention to retrieve relevant information from a visual or textual KB, which is in turn accumulated into memory passed from one iteration to the next. However, in contrast to our model, it views the question as an atomic unit, whereas our model decomposes it into a multi-step action plan informing each cell in our sequential network about its current objective. Another key difference is the distant interaction between the question and the KB that characterizes our model. Conversely, DMN fuses their corresponding representations together into the same vector space.

Graves et al. (2016; 2014) complements a neural network with a memory array it can interact with, through the means of soft attention. Analogously to our model, it partitions the model into a core neural network, called controller, as well as reading and writing heads that interact with external memory array. However, a main point distinguishing our model from this approach, is the use of dynamic memory, as in Kumar et al. (2016), instead of a fixed-array memory. Each MAC cell is associated with a memory state, our reading unit inspects only the latest memory passed from the previous state, and our writing unit creates a new memory state rather than writing to multiple slots in a fixed shared external memory. Notably, our approach is much more reminiscent of the widely successful RNN structure, rather than to Graves et al. (2016; 2014) .

Finally, our approach has potential ties to the VQA models Hu et al. (2017); Lu et al. (2016) which also attend both the to question words and the image while progressively addressing the given question. However, both of these models have distinct specialized designs for each of their attention layers or modules, and have a discrete or fixed layout in which they are composed together. In contrast, our approach relax both of these limitations, having one universal cell design and one universal self-attending sequential network layout.

## C.4 ATTENTION VS. CONVOLUTION

Compared to other leading methods, our model stands out by being heavily based on soft attention, whereas most competing approaches are CNN-based, surprisingly lack any attention mechanism. Since attention is commonly used in models designed for standard VQA (Antol et al., 2015; Gupta, 2017; Lu et al., 2016; Yang et al., 2016), it is reasonable to assume that it would be beneficial to incorporate such methods into visual reasoning systems for the CLEVR task as well. In fact, attention mechanisms should be especially useful for multi-step reasoning questions such as those present in CLEVR. Such questions refer to several relations between different objects in the image and feature compositional structure that may be approached one step at a time. Thus, it should be beneficial for a cogent responder to have the capacity to selectively focus on on one or some objects at each step, traversing the relevant relational links one after the other, both at the image level, and at the question level.

Moreover, attention mechanisms enhance our model's ability to perform reasoning skills that pertain to aggregation of information across different regions, such as counting, finding maximum value, or performing other reduction operations over information that is spread across the image. Indeed, as discussed in 4, all existing models for visual reasoning, most of which lacking any attention mechanism, struggle with the counting and numerical comparisons questions present in CLEVR. Conversely, our model proves much more capable of performing these reasoning skills, outperforming the other approaches by a wide margin. Noticeably, incorporating soft attention into our model makes it much more adept at performing such aggregation reasoning skills, successfully addressing the this type of questions.

Finally, as pointed out by Lu et al. (2016); Yang et al. (2016), soft attention confers the model with robustness to noise introduced from irrelevant information presents in the image, and higher capacity for handling larger and more diverse vocabulary, the latter being demonstrated in 4. It allows the model to separate the wheat from the chaff, selectively attending to the relevant information only, and arguably, being more resilient to both visual and linguistic variations.

