# OpenReview forum: "Compositional Attention Networks for Machine Reasoning"
_ICLR.cc/2018/Conference — Accept (Poster)_

### Official Review · AnonReviewer2 · 2017-11-27

**Rating:** 7
**Confidence:** 3

**Review:**

This paper describes a new model architecture for machine reasoning. In contrast
to previous approaches that explicitly predict a question-specific module
network layout, the current paper introduces a monolithic feedforward network
with iterated rounds of attention and memory. On a few variants of the CLEVR
dataset, it outperforms both discrete modular approaches, existing iterated
attention models, and the conditional-normalization-based FiLM model.

So many models are close to perfect accuracy on the standard CLEVR dataset that
I'm not sure how interesting these results are. In this respect I think the
current paper's results on CLEVR-Humans and smaller fractions of synthetic CLEVR
are much more exciting.

On the whole I think this is a strong paper. I have two main concerns. The
largest is that this paper offers very little in the way of analysis. The model
is structurally quite similar to a stacked attention network or a particular
fixed arrangement of attentive N2NMN modules, and it's not at all clear based on
the limited set of experimental results where the improvements are actually
coming from. It's also possible that many of the proposed changes are
complementary to NMN- or CBN-type models, and it would be nice to know if this
is the case.

Secondarily, the paper asserts that "our architecture can handle
datasets more diverse than CLEVR", but runs no experiments to validate this. It
seems like once all the pieces are in place it should be very easy to get
numbers on VQA or even a more interesting synthetic dataset like NLVR.

Based on a sibling comment, it seems that there may also be some problems with
the comparison to FiLM, and I would like to see this addressed.

On the whole, the results are probably strong enough on their own to justify
admitting this paper. But I will become much more enthusiastic about if if the
authors can provide results on other datasets (even if they're not
state-of-the-art!) as well as evidence for the following:

1. Does the control mechanism attend to reasonable parts of the sentence?

Here it's probably enough to generate a bunch of examples showing sentence
attentions evolving over time.

2. Do these induce reasonable attentions over regions of the image?

Again, examples are fine.

3. Do the self-attention and gating mechanisms recover the right structure?

In addition to examples, here I think there are some useful qualitative
measures. It should be possible to extract reasonable discretized "reasoning
maps" by running MST or just thesholding on the "edge weights" induced by
attention and gating. Having extracted these from a bunch of examples, you can
compare them to the structural properties of the ground-truth CLEVR network
layouts by plotting a comparison of sizes, branching factors, etc.

4. More on the left side of the dataset size / accuracy curve. What happens if
   you only give the model 7000 examples? 700? 70?

Fussy typographical notes:

- This paper makes use of a lot of multi-letter names in mathmode. These are
  currently written like $KB$, which looks bad, and should instead be
  $\mathit{KB}$.

- Variables with both superscripts and subscripts have the superscripts pushed
  off to the right; I think you're writing these like $b_5 ^d$ but they should
  just be $b_5^d$ (no space).

- Number equations and then don't bother carrying subscripts like $W_3$, $W_4$
  around across different parts of the model---this isn't helpful.

- The superscripts indicating the dimensions of parameter matrices and vectors
  are quite helpful, but don't seem to be explained anywhere in the text. I
  think the notation $W^{(d \times d)}$ is more standard than $W^{d, d}$.

- Put the cell diagrams right next to the body text that describes them (maybe even
  inline, rather than in figures). It's annoying to flip back and forth.

---

> ### Comment · AnonReviewer2 · 2018-01-11
> **Response to rebuttal**
>
> Thanks for the ablations! My score remains the same.

---

> > ### Author Response · Authors · 2018-01-12
> > **To Reviewer2:**
> >
> > Alright. Thank you very much for your detailed review - it was very helpful to us and we truly appreciate it! We are currently working on Cornell nlvr and also text-based datasets as well as more qualitative experiments and while I believe I shouldn't upload revisions anymore we will definitely explore these directions further!

---

### Official Review · AnonReviewer1 · 2017-11-27
**Looks good but needs clarification**

**Rating:** 6
**Confidence:** 4

**Review:**

This paper proposes a recurrent neural network for visual question answering. The recurrent neural network is equipped with a carefully designed recurrent unit called MAC (Memory, Attention and Control) cell, which encourages sequential reasoning by restraining interaction between inputs and its hidden states. The proposed model shows the state-of-the-art performance on CLEVR and CLEVR-Humans dataset, which are standard benchmarks for visual reasoning problem. Additional experiments with limited training data shows the data efficiency of the model, which supports its strong generalization ability.

The proposed model in this paper is designed with reasonable motivations and shows strong experimental results in terms of overall accuracy and the data efficiency. However, an issue in the writing, usage of external component and lack of experimental justification of the design choices hinder the clear understanding of the proposed model.

An issue in the writing
Overall, the paper is well written and easy to understand, but Section 3.2.3 (The Write Unit) has contradictory statements about their implementation. Specifically, they proposed three different ways to update the memory (simple update, self attention and memory gate), but it is not clear which method is used in the end.

Usage of external component
The proposed model uses pretrained word vectors called GloVE, which has boosted the performance on visual question answering. This experimental setting makes fair comparison with the previous works difficult as the pre-trained word vectors are not used for the previous works. To isolate the strength of the proposed reasoning module, I ask to provide experiments without pretrained word vectors.

Lack of experimental justification of the design choices
The proposed recurrent unit contains various design choices such as separation of three different units (control unit, read unit and memory unit), attention based input processing and different memory updates stem from different motivations. However, these design choices are not justified well because there is neither ablation study nor visualization of internal states. Any analysis or empirical study on these design choices is necessary to understand the characteristics of the model. Here, I suggest to provide few visualizations of attention weights and ablation study that could support indispensability of the design choices.

---

> ### Author Response · Authors · 2018-01-16
> **Dear Reviewer1:**
>
>
> Thank you very much for your review - we truly appreciate it!
> We have uploaded a revision (by the rebuttal deadline, jan 5) that addresses all your comments:
>
> 1. We have revised the description of the writing unit to make it more clear - we have experimented with several variants for this unit - the "standard" one (for which all the results are about), and 3 variants: a. with self-attention, b. with gating, and c. with both self-attention and gating. In the ablations study section we have included results for each of these for the whole dataset, 10% of the dataset and also showed training curves for each variant.
>
> 2. We have trained the models without GloVE and added these results along with clarification to the experiments section.
>
> 3. We have included ablation studies in order to justify the architecture design choices and elucidate their impact. We have also added visualizations of attention weights for several examples and discussed them.
>
> Thanks a lot again for your review!
> - Paper858 Authors

---

### Official Review · AnonReviewer3 · 2017-11-28
**Needs more analysis and qualitative evaluation**

**Rating:** 7
**Confidence:** 4

**Review:**

Summary:
The paper presents a new model called Compositional Attention Networks (CAN) for visual reasoning. The complete model consists of an input unit, a sequence of the proposed Memory, Attention and Composition (MAC) cell, and an output unit. Experiments on CLEVR dataset shows that the proposed model outperforms previous models.

Strengths:
— The idea of building a compositional model for visual reasoning and visual question answering makes a lot of sense, and, I think, is the correct direction to go forward in these fields.
— The proposed model outperforms existing models pushing the state-of-the-art.
— The proposed model is computationally cheaper and generalizes well with less training data as compared to existing models.
— The proposed model has been described in detail in the paper.

Weaknesses:
— Given that the performance of state-on-art on CLEVR dataset is already very high ( <5% error) and the performance numbers of the proposed model are not very far from the previous models, it is very important to report the variance in accuracies along with the mean accuracies to determine if the performance of the proposed model is statistically significantly better than the previous models.
— It is not clear which part of the proposed model leads to how much improvement in performance. Ablations studies are needed to justify the motivations for each of the components of the proposed model.
— Analysis of qualitative results (including attention maps, gate values, etc.) is needed to justify if the model is actually doing what the authors think it should do. For example, the authors mention an example on page 6 at the end of Section 3.2.2, but do not justify if this is actually what the model is doing.
— Why is it necessary to use both question and memory information to answer the question even when the question was already used to compute the memory information? I would think that including the question information helps in learning the language priors in the dataset. Have the authors looked at some qualitative examples where the model which only uses memory information gives an incorrect answer but adding the question information results in a correct answer?
— Details such as using Glove word embeddings are important and can affect the performance of models significantly. Therefore, they should be clearly mentioned in the main paper while comparing with other models which do not use them.
— The comparisons of number of epochs required for training and the training time need fixed batch sizes and CPU/GPU configurations. Is that true? These should be reported in this section.
— The authors claim that their model is robust to linguistic variations and diverse vocabulary, by which I am guessing they are referring to experiments on CLEVR-Humans dataset. What is there in the architecture of the proposed model which provides this ability? If it is the Glove vectors, it should be clearly mentioned since any other model using Glove vectors should have this ability.
— On page 6, second paragraph, the authors mention that there are cases which necessitate the model to ignore current memories. Can the authors show some qualitative examples for such cases?
— In the intro, the authors claim that their proposed cell encourages transparency. But, the design of their cell doesn’t seem to do so, nor it is justified in the paper.

Overall: The performance reported in the paper is impressive and outperforms previous state-of-the-art, but without proper statistical significance analysis of performance, ablation studies, analysis of various attention maps, memory gates, etc. and qualitative results, I am not sure if this work would be directly useful for the research community.

---

> ### Comment · AnonReviewer3 · 2018-01-16
> **Final recommendation**
>
> Thanks for the statistical significance analysis, ablation studies, qualitative results and other clarifications, I have increased my rating from 6 to 7. Please fix minor things such as 2 (different) plots for Network Length in Figure 7, short explanation for why the attention is significant on the word "is" instead of an important word "shape" in qualitative examples 1&2 in figure 8, etc. Also, the current paper length is 14 pages (after the addition of about 4 pages in rebuttal) which is almost double the recommended length of 8 pages, so I would suggest reducing the paper length for future.

---

> > ### Author Response · Authors · 2018-01-16
> > **Thank you!**
> >
> > Thank you very much for your review and for improving the rating! We will fix the typos and indeed are also working on making the writing a bit more concise to shorten the overall paper length!

---

### Public Comment · ~Ethan_Perez1 · 2017-11-22
**Questions about comparisons to competitors and why CAN works**

Congratulations on the impressive results. The proposed CAN model is quite interesting. We are also grateful that you represented our work with FiLM respectfully :)

We have a few questions:
1) How does CAN perform when learning word embeddings from scratch, as competing models do, rather than using GloVE embeddings? It seems that using external knowledge via pre-trained embeddings would lead to unfair comparisons for:
  --CLEVR-Humans performance: CLEVR-Humans has many new words, which competing methods learn from scratch from a small training set or else use <UNK> tokens.
  --CLEVR training curves: External word knowledge frees a model from having to learn word meanings from scratch early in training, facilitating high performance early on.
  --CLEVR performance (to a lesser extent): GloVE might encode helpful notions, not learnable from CLEVR, on how to reason about various words.
  --It seems that results using GloVE should have an accompanying asterisk denoting so in tables and that GloVE’s use should be noted in the main paper body (not just appendix). Even better, would it be too difficult to re-run these CAN experiments with learned word embeddings?
2) How did you run competitor models? There are discrepancies which make us hesitant to accept the self-run competitors results that the paper reports and compares against:
  --“Training curve” figure: Here are the FiLM validation accuracies for the reported model for first 11 epochs (plotted in (Perez et al. 2017)), higher than your paper plots: [0.5164, 0.7251, 0.802, 0.8327, 0.8568, 0.8887, 0.9063, 0.9243, 0.9328, 0.9346, 0.942]
  --“Accuracy / Dataset size (out of 700k)” figure: Trained on 100% of CLEVR, FiLM achieves 97.84% validation acc., as reported in our paper (Perez et al. 2017), compared to ~94% in your plot. We have also run incomplete FiLM experiments on 25% and 50% subsets of CLEVR and achieved higher numbers than those you report/plot. We can run these full experiments and comment numbers if that would be helpful.
  --“Accuracy / Dataset size (out of 700k)” figure: You plot that PG+EE achieves ~95.5% validation acc., lower than the reported 96.9% test acc. (a likely lower bound for validation acc.).
3) Do you have evidence supporting the reason(s) behind CAN’s success? The paper only gives 2 pages of experiments and analysis, all quantitative and related to outperforming other models, rather than qualitative, ablative, or analytical. Thus, it’s difficult to tell which of the paper’s several proposed intuitions on why CAN works so well are valid. CAN consists of many components and aspects (control/read/write units, memory gates, compositionality, sequential reasoning, parameter-sharing, spatial attention, self-attention, etc.), and it’s unclear from overall acc. numbers which ones are crucial. In particular:
  --Is there evidence spatial attention allows CAN to reason about space more effectively? Intuitively, spatial attention should help, but (Johnson et al., 2016; Santoro et al., 2017) show that spatial attention models struggle with CLEVR/spatial reasoning; in the extreme, it’s possible that CAN performs well in spite of spatial attention. On the other hand, (Perez et al. 2017) show, with CLEVR acc. and activation visualizations, that an attention-free method (FiLM) can effectively reason about space. Can you run analytical experiments or visualizations to support that spatial attention actually helps CAN reason about space? Where is the model attending after each reasoning step? What is the CLEVR acc. after replacing spatial attention with another form of conditioning? What errors does CAN make and not make, especially to other models? If it would help, we can provide our reported FiLM model via GitHub.
  --It seems possible to build high-level aspects of CAN into simpler reasoning architectures and achieve similar performance. I.e., if spatial attention is key, adding spatial attention layers to a FiLM-based model, as in [1], might recover CAN’s performance. We would be interested to see evidence (such as ablations) showing that the entire CAN model is necessary for successful reasoning, as claimed.
  --How does question-attention vary across reasoning steps? Does CAN iteratively focus on the query’s various parts, as claimed? Or does CAN’s attention not hop around, entirely ignore some question parts, etc.? CAN’s attention lends itself to very neat analysis possibilities.
  --Do simpler questions actually use the memory gate to shorten the number of reasoning steps as expected?

Overall, given the strong numerical results, CAN seems to potentially be a quite promising model, pending the authors’ response to these questions and some further analysis and evidence.

Kind Regards,
Ethan Perez, Florian Strub, Harm de Vries, Vincent Dumoulin, Aaron Courville

References:
[1] Modulating early visual processing by language. Harm de Vries, Florian Strub, Jérémie Mary, Hugo Larochelle, Olivier Pietquin, Aaron Courville. NIPS 2017.

---

> ### Author Response · Authors · 2017-11-23
> **Re: Questions about comparisons to competitors and why CAN works (Part 2)**
>
> 3) Qualitative Experiments
>
> While developing our idea, we have performed a large number of experiments that test ablations, modifications and variations to our architecture, the results of which will be presented in a few days in the revised version of paper (once it becomes possible to submit revisions). These experiments indeed demonstrate the relative importance of each of the model’s different aspects to its overall performance. For instance, they show that while optional components of the Write Unit such as memory gating and self-attention (discussed in section 3.2.2) improve the final performance and accelerate training, the model still achieves strong state-of-the-art results without them. Conversely, they quantitatively show the importance of using attention to decompose the question into a series of control states, in contrast to having only spatial attention layers over the image, as is the case for example in stacked attention networks (Yang et al., 2016; Johnson et al., 2017). In addition, we have examined the performance of the model across different network lengths (i.e. number of MAC cells), hidden state dimensions, and with several types of nonlinearities, all are establishing the robustness of our model to implementation details and model variances.
>
> We are actively exploring our model behavior in qualitative terms and, in the paper revision, we will show gate-values and attention-map visualizations over the image, question and previous memories, following each reasoning step and for different types of questions. Indeed, soft-attention lies at the heart of our model, conferring several advantages, as discussed in the paper: (1) Robustness against both linguistic and visual variations (corroborated also by Lu et al (2016) and Yang et al (2016)). (2) Capacity for easy-to-train multi-modal translation between attended question words to the corresponding attended regions in the image. (3) Compositionality of reasoning steps by standardizing the content of the dual control and memory states to be selective summaries of the question and image correspondingly, and finally, as you alluded to, (4) Interpretability - Ideally, the model may be able to show a clear step-by-step rationale supporting the predicted answer.
>
> Furthermore, we are working on an error analysis for both CLEVR and CLEVR-Humans datasets to have indeed a better understanding of the nature of mistakes our model makes. Based on the quantitative results in the paper, it is already noticeable that many of the errors are for the counting questions, which is reasonable given the larger output space they have compared to other question types. We are currently looking into potential ways for further improving models counting accuracies, and will add a discussion about that either in a revision of the paper or in future work.
>
> --
>
> Thank you very much for the thorough and insightful response and suggestions for further exploration! We will upload a revised version with the aforementioned additions once the option becomes available! :)
> - Paper858 Authors

---

> ### Author Response · Authors · 2017-11-23
> **Re: Questions about comparisons to competitors and why CAN works (Part 1)**
>
> Thank you very much for the kind words and for the detailed response! We truly appreciate it!
>
> Addressing the questions you have raised:
>
> 1) GloVE Word Embeddings
>
> For the CLEVR dataset, we have observed an improvement of 0.17% in the final validation accuracy when using GloVE compared to word vectors initialized randomly with standard normal distribution, and 0.24% improvement compared to uniform-distribution initialization with range [-1,1].
>
> Notably, in the early stages of the training process, the models with learned-from-scratch word embeddings actually outperformed the model with pretrained GloVE word embeddings. Only by the end of the training this trend is reversed to a modest advantage for the GloVE-based model. These results, which we will be happy to add to the paper, suggest that randomly-initialized word vectors are in fact easier for the model to distinguish between initially, whereas the small advantage of some additional semantic notions embodied in the GloVE embeddings become noticeable only by the end of training, useful for a low fraction of the questions. In any case, we will be glad to stress the fact that we have used GloVE in the experiments section of the paper.
>
> For CLEVR-Humans, we so far have indeed used the pre-trained vectors and, as mentioned in the paper, haven’t trained them any further to prevent a drift in their semantic meaning. I agree that it will be both interesting and fair to check the model performance on CLEVR-Humans for randomly-initialized vectors as well. I will be running an experiment for that right now, and so we will report the scores as soon as they arrive and update the paper with these additional results accordingly.
>
> 2) Comparison to other models
>
> For the competitor models, we have used the original publicly available implementations as-is for both FiLM (https://github.com/ethanjperez/film) and PG+EE (https://github.com/facebookresearch/clevr-iep) with their default arguments, after closely following all the training procedures listed on the websites (image features extraction, question preprocessing etc.).
>
> For figure 4 Right, while indeed there is some difference between the numbers you report and the numbers we have obtained by self-running the mentioned github version, when compared to performance of other approaches, the difference seems to be quite modest in relative terms. It may be a good idea to run FiLM and other models several times and measure both the average and variance in performance across these attempts. I think some variance between two different runs of the same model can be generally expected even given the equal settings, and this also depends on the model stability and robustness. In any case, as shown in figure 4, the gap in performance between CANs and other models is consistently significant throughout the training process, and it remains the case also for the results you mention.
>
> For figure 4 Left, accuracy as a function of the training-set size, it may be the case that the discrepancy you claim arises from difference in the training time that has been allowed before collecting the results: with the aim of having a fair comparison, we have run all the models the same amount of time: until the validation score of all models has not shown further improvement for several consecutive iterations. It may be the case that allowing longer time for the training of FiLM and other models will lead to better scores, and we will be happy to mention them as well in the paper. Indeed, there are trade-offs between training time, dataset size and accuracy that are interesting to explore.
>
> Another important aspect pertaining to the comparability of different models is their size - the number of parameters being used. We have noted that FiLM has a relatively high dimension of 4096 for the  question-processing GRU hidden states, especially when compared to competing approaches that use sizes of 128-512 (our model has hidden dimension of 512, achieving similar results for 256 after slightly longer training time). Since the size of the weight matrix used in the GRU is quadratic as a function of the hidden state size, this leads to O(4096*4096) parameters which is, by and large, an order-of-magnitude higher than O(512*512) or O(256*256). In order to have a more fair comparison, it seems that it may be interesting to test FiLM with a smaller state size, comparably to other models.

---

### Author Response · Authors · 2017-12-13
**Re: the reviews**


Dear reviewers,

Thank you very much for your detailed comments and insightful suggestions for further exploration. We completely agree that ablation studies, statistical significance measures and qualitative analysis such as visualizations are necessary to justify the performance of the model and elucidate its behavior. We are actively working on a revised version of the paper that will include these studies and address the other comments raised in the reviews, in particular regarding the use of GloVE, the model’s performance on smaller subsets (<10%) of CLEVR, and the necessity of predicting the answer using both the question and the memory.

Several clarifications in response to questions from the reviews:

1. For comparative experiments, all the other systems used the original publicly-available authors’ implementations. All the models were trained with an equal batch size of 64 (as in the original implementations) and on the same machine, using a single Titan X Maxwell GPU per model.

2. As mentioned in section 3.2.4, our claims about the model’s robustness to linguistic variability and its ability to handle datasets more diverse than the standard CLEVR indeed refer to its performance on CLEVR-Humans. We believe that attention mechanisms used over the question are a key factor that allows that, as discussed in the supplementary material, section C.4, and supported by Lu et al. (2016) and Yang et al. (2016). We will address this matter further in the revised version of the paper.

Thank you!
- Paper858 Authors

---

### Author Response · Authors · 2018-01-05
**Revision**


Dear reviewers,

Thank you very much for your detailed suggestions and comments!
We have uploaded a revision that addresses the comments raised in the reviews. In particular:
- We report the model performance with random initialization for the word vectors rather than GloVE embeddings. Using a uniform initialization, we were able to get equivalent results for CLEVR and 1-2% difference for CLEVR-humans in the new setting, compared to those we have achieved with GloVE.
- We have performed statistical significance analysis, running each model multiple times (10) and updated the plots to show averages and confidence intervals. These experiments show that the results of our model are indeed statistically significant compared to the alternative models.
- We have performed ablation studies that cover the control, reading, and writing units as well as additional aspects of our model. We have shown results for the standard CLEVR dataset and 10% subset of it along with training curves. The ablation study we have conducted quantifies the contribution of each of the model components to its overall performance and shows its relative significance. Based on the results, we can see the importance of using attention over both the question and the image in a coordinated manner through a dual structure of recurrent control and memory paths.
- We have looked into attention maps of the model for the question and image and put a few examples of these in the revised version of the paper that demonstrate the model interpretability.
- For the gating mechanism of the writing unit, we have performed additional experiments showing that untied gate values for each entry of the state vector perform better than having one shared potentially-interpretable gate for the whole state and so have changed the description of that subsection accordingly.

Additional changes in the paper:
- We fixed typos that were in the original submission.
- We have clarified a few missing points that were mentioned in the reviews.
- This includes in particular clarification about the writing unit mods of operation, and the several variants - with self-attention, with a gate, and with both. Each of these variants is accompanied with training curves and final accuracies as part of the ablation section.

In response to specific comments by reviewers:
- The ablations study shows the importance of using both the question and the final memory state. As explained in the revision, since the memory holds information only from the image, it may not contain all required information to answer the question, since potentially crucial aspects of the question are not represented in the image.
For instance: given an image with one object, one question can ask about its size and another question about its color, but in both cases the memory will attend to the same one object and thus will not contain enough information to respond to the question correctly.
- As supported by both the ablations studies and the new experiments that show model's performance on CLEVR-humans without using GloVE, we claim that the model owes its ability to handle diverse language and learn from small amounts of data to the use of attention over question. In particular, the attention allows the model to ignore varied words in the CLEVR-humans dataset that the model hasn't been necessarily trained on but aren't crucial to understand the question, and rather can focus only on the key words that refer to the objects and their properties.
- As demonstrated by the qualitative results of attention maps over the image and question and explained in the model section, the model is indeed transparent by having access to the attention maps of each computation step. And indeed, examples of such attention maps show interpretable rationales behind the model's predictions.

Thank you very much!
- Paper858 Authors

---

### Author Response · Authors · 2018-04-14
**Code and updated paper**

Updated version of the paper is at: https://arxiv.org/abs/1803.03067
Implementation of the model is at: https://github.com/stanfordnlp/mac-network
See https://cs.stanford.edu/people/dorarad/mac/ for further information and updates!

Thank you!

---

### Decision · Program_Chairs · 2018-01-29
**ICLR 2018 Conference Acceptance Decision**

**Decision:**

Accept (Poster)

**Comment:**

PROS:
1. Good results on CLEVER datasets
2. Writing is clear
3. The MAC unit is novel and interesting.
4. Ablation experiments are helpful

CONS:
The authors overstate the degree to which they are doing "sound" and "transparent" reasoning.  In particular statements such as "Most neural networks are essentially very large correlation engines that will hone in on any statistical, potentially spurious pattern that allows them to model the observed data more accurately. In contrast, we seek to create a model structure that requires combining sound inference steps to solve a problem instance." I think are not supported.  As far as I can tell, the authors' do not show that the steps of these solutions are really doing inference in any sound way

I also found the interpretability section to be a bit unconvincing.  The reviewers and I discussed this and there was some attempt to assess what the operations were actually doing but it is not clear how the language and the image attention are linked.

I wonder whether the learned control activations are abstract and re-used across problems the way that the accompanying functional solution's primitives are.  Have you looked at how similar the controls are across problems which are identical except for a different choice of attributes?  To me, one of the hallmarks of a truly "compositional" solution is one in which the pieces are re-used across problems, not just that there is some sequence of explicit control activations used to solve each individual problem.